# Pacing in Long-Distance Running: Sex and Age Differences in 10-km Race and Marathon

**DOI:** 10.3390/medicina57040389

**Published:** 2021-04-17

**Authors:** Ivan Cuk, Pantelis T. Nikolaidis, Elias Villiger, Beat Knechtle

**Affiliations:** 1Faculty of Physical Education and Sports Management, Singidunum University, 11000 Belgrade, Serbia; ivan_cuk84@yahoo.com; 2Exercise Physiology Laboratory, 18450 Nikaia, Greece; pademil@hotmail.com; 3School of Health and Caring Sciences, University of West Attica, 10679 Athens, Greece; 4Institute of Primary Care, University of Zurich, 8006 Zurich, Switzerland; eviliger@gmail.com; 5Medbase St. Gallen Am Vadianplatz, 9000 St. Gallen, Switzerland

**Keywords:** running, endurance, health, marathoners, recreation

## Abstract

*Background and objective*: The recent availability of data from mass-participation running events has allowed researchers to examine pacing from the perspective of non-elite distance runners. Based on an extensive analysis of the literature, we concluded that no study utilizing mass-participation events data has ever directly compared pacing in the 10-km race, with other long-distance races. Therefore, the main aim of this study was to assess and compare pacing between 10-km runners and marathoners, in regards to their sex and age. *Materials and methods*: For the purpose of this study, official results from the Oslo marathon (*n* = 8828) and 10-km race (*n* = 16,315) held from 2015 to 2018 were included. *Results*: Both 10-km runners and marathoners showed positive pacing strategies. Moreover, two-way analysis of variance showed that women were less likely to slow in the marathon than men (9.85% in comparison to 12.70%) however, not in the 10-km race (3.99% in comparison to 3.38%). Finally, pace changing is more prominent in youngest and oldest marathoners comparing to the other age groups (12.55% in comparison to 10.96%). *Conclusions*: Based on these findings, practitioners should adopt different training programmes for marathoners in comparison to shorter long-distance runners.

## 1. Introduction

Pacing can be defined as the distribution of exercise intensity during a prolonged time [1]. Optimal pacing is one of the most important contributors to achieving the best results in long-distance running [1,2], while significantly decreasing the risk of musculoskeletal injuries [3]. Although a pacing strategy that can classify as optimal depends on many factors (e.g., race length and profile, altitude, or weather conditions [4,5]) an even pacing strategy with an end-spurt has often been the best choice for long-distance events [6]. This strategy was best seen in the recent successful sub-2-h marathon challenge by Eliud Kipchoge, where the pacing was artificially controlled to be even throughout the race, with the spontaneous end-spurt by Kipchoge in the last several hundred meters [7]. Different pacing strategies in long-distance running were extensively investigated by several influential studies, both on track [4] and road racing [8,9]. However, those studies only involved a small sample size of professional athletes. On the other hand, the recent availability of mass-participation events data has allowed researchers to examine pacing from the perspective of a wider range of athletes including recreational distance runners of all ages [10,11]. 

The first studies using mass-participation events data were focused on independent long-distance events, primarily marathons and half-marathons. Contrary to elite runners, when a wider range of athletes was investigated, previous research reported positive pacing in endurance running races, where pace (time/distance) increased after approximately ¾ of the race [12,13]. This decline in running speed was more prominent in men than in women due to several physiological [12] and psychological factors [14,15], such as a riskier and faster start by men or women’s better fat utilization to obtain energy.

Moreover, studies investigating pacing in age group endurance runners proved to be somewhat inconsistent in their findings, with either no differences between age groups [11,12] or with more even pacing in older age groups [13]. Regarding the 10-km race, several studies examined pacing in this long-distance event, mainly in a small samples of track runners [1,4], time trial runners on either track [16] or treadmill [17], as well as in triathletes [18]. Pacing in the aforementioned elite 10-km runners proved to be rather even with an end-spurt. However, in recent years, 10-km races became increasingly popular in recreational runners, especially among women and beginners of both sexes [19,20]. Recreational athletes would participate in a 10-km race in the context of their preparation for a subsequent longer race [19], as well as a training tool to ameliorate pace time [21]. Therefore, further investigation of pacing in recreational 10-km runners is crucial to better understand the mechanisms controlling the pacing in long-distance events. This might help runners to enjoy running more as well as to achieve better results. 

In recent years, several studies using new methodological approaches attempted to compare pacing in mass-participation events [10,20]. Nevertheless, the performance of different distances and events was not adjusted in the abovementioned studies; e.g., some events might be under different environmental conditions, opposition fields, or race profiles. On the other hand, a new methodological approach allowed researchers to directly compare pacing between half-marathon and marathon on the same race and track, with rather similar weather conditions [10,22], thus providing more detailed and comparable results. For example, a novel finding was that women’s pacing was similar to men’s in half-marathon, whereas in a marathon women had more even pacing compared to men. Specifically, physiological rather than psychological factors can influence the additional lack of speed in marathoners (and not half-marathoners), such as better utilization of fat by women or men’s muscle glycogen depletion [10,22]. Accordingly, women did not differ by age group in pace variability, whereas youngest and oldest men, showed larger variability in pace [10]. However, further proof is needed that the observed sex and age differences are not specific to only a few races (i.e., Vienna and Ljubljana) and only two long-distance running events (i.e., half-marathon and marathon). This can be achieved by investigating and comparing pacing strategies in other long-distance races, such as longer ultra-races, or shorter and increasingly popular 10km races with already popular and investigated half-marathon or marathon. 

Based on an extensive analysis of the literature, we concluded that no study utilizing mass-participation events data has ever directly compared pacing in a 10-km race, with other long-distance races. Such a comparison could shed additional light on the importance of the mechanisms underlying pacing behaviour of the long distance runners, as well as to better understand potential training requirements for both recreational and proficient runners. Therefore, the main aim of this study was to assess and compare pacing between the increasingly popular 10-km race and the most popular long-distance race—marathon, in regards to their sex and age. We hypothesized that 10-km runners will show more even pacing than marathoners, particularly women and middle age runners. 

## 2. Materials and Methods

For this study, official results from the Oslo marathon and Oslo 10-km race held from 2015 to 2018 were included [23]. Split times from the middle of the race were also included (i.e., 5 km and 21.0975 km for the 10-km race and marathon respectively).

The Oslo marathon was chosen as an officially certified race because it was held on a rather flat track (elevation difference 60 m). For reference, the Berlin Marathon considered “the fastest marathon”, has an elevation difference of 21 m [24]. Moreover, both the 10-km race and marathon were held on the same day, whereas the 10-km race was entirely contained within the marathon race. Finally, note that hyperthermia can significantly affect pacing in both elite and recreational runners [5,12]. The Oslo Marathon is traditionally held in Norway at the end of September, usually in colder weather conditions, which can reduce the chances of hyperthermia in runners.

### 2.1. Participants

In total, 25,143 participants of all performance levels were considered for this study (10-km race, *n* = 16,315; Marathon, *n* = 8828), however, most of them were recreational runners. Participants who did not finish any of the races, or did not have recorded any of the split times were excluded from the initial sample.

The present research was approved by the Institutional Review Board of Kanton St. Gallen, Switzerland, with a waiver of the requirement for informed consent of the participants as the research concerned the study of publicly available data (Ethical Committee St. Gallen 1 June 2010). This research was conducted in accordance with ethical standards derived from the Declaration of Helsinki adopted in 1964 and revised in 2013.

### 2.2. Data Acquisition

All data was acquired from the official Oslo Marathon results page [23]. First, overall times, athlete details and a link to each athlete’s split times were copied from the results page and pasted into an Excel document. This was done separately per year, distance and gender. The split times were then added in a second step using custom Python scripts that followed the official link to each athlete’s split times and extracted all available split times.

### 2.3. Procedures

In the first step of data analysis, the average running speed of all runners was calculated for the first and second half of both the 10-km race and marathon. A particular novelty of this study was the use of time (i.e., minutes and seconds) per kilometre as a unit of speed. This “runners friendly” measurement of speed was chosen as a very practical tool for both professional and recreational runners as well as their coaches. For example, GPS watches, often utilized by runners to monitor running speed, presents minutes per kilometre by default. Moreover, in long-distance races, each kilometre is usually marked. Therefore, participants can see the time they consumed running between each kilometre. As a result, runners and running coaches often rely on the time needed to run one kilometre when assessing and comparing someone’s running speed or pace maintenance.

Considering that even pacing is the best choice for long-distance running [6], pacing assessment from the aspect of speed maintenance was selected for this study. Thereafter, speed variation was calculated based on the percentage difference in speed observed between the second and the first half of the race (i.e., % change = (second half time − first half time)/first half time). Percentage change was considered as a continuous variable [14]. Applying this method, it was possible to normalize pace and compare it between different race distances [10]. Criteria for inclusion in the final data set were having timing data for the halfway mark and the full race in proper sequence (e.g., finishing time greater than split time); a net time less than the gun time; and a slowing less than 400% [14].

### 2.4. Statistical Analysis

Prior to all statistical tests, descriptive statistics were calculated as mean and standard deviation. Since the Kolmogorov-Smirnov or similar data normality tests are not sensitive when using a large sample size, data distribution normality was verified by visual inspection of histograms and QQ plots [10,22]. 

To assess pacing differences between the first and second half of the 10-km race and marathon, two 2-way between-within ANOVAs were performed (separately for women and men). The main effect of pace (first half and second half), race (10-km race and marathon), and the interaction pace x race were observed.

To assess pace change between women and men in 10-km race and marathon, two-way ANOVA with between factors was performed. The main effect of sex (women and men), race (10-km race and marathon), and the interaction sex × race were observed.

Finally, to assess pace change between age groups in 10-km race and marathon, two 2-way between-within ANOVAs were performed (separately for women and men). The main effect of age group (18–23; 24–34; 34–39; 40–44; 45–49; 50–54; 55–59; 60–64; 65+), race (10-km race and marathon) and the interaction age × race were observed. 

In addition, for all ANOVAs, Bonferroni post-hoc test was performed. The effect size was calculated as eta squared (ŋ^2^), where the values of 0.01, 0.06, and above 0.14 were considered small, medium, and large, respectively [25]. Alpha level was set at *p* ≤ 0.05. All statistical tests were performed using Microsoft Office Excel 2007 (Microsoft Corporation, Redmond, WA, USA) and SPSS 20 (IBM, Armonk, NY, USA).

## 3. Results

The first and second half pacing of participants is presented in Table 1. Regardless of their sex and age, both 10-km runners and marathoners showed a positive pacing strategy (i.e., second half of the race was slower than the first half). Further examination of pacing between 10-km runners and marathoners, in regards to their sex and age, is presented in Figure 1, Figure 2 and Figure 3.

### 3.1. Pacing in 10-km and Marathon

For women (Figure 1, upper panel), the two-way ANOVA showed significant main effects of *pace*(F_(3,11978)_ = 6513.1, ŋ^2^ = 0.02, *p* < 0.01), *race*(F_(3,11978)_ = 187.7, ŋ^2^ = 0.01, *p* < 0.01) as well as *pace × race*interaction (F_(3,11978)_ = 1086.1, ŋ^2^ < 0.01, *p* < 0.01), whereas for men (Figure 1, lower panel), the two-way ANOVA showed significant main effects of *pace*(F_(3,13161)_ = 8720.9, ŋ^2^ = 0.04, *p* < 0.01), *race*(F_(3,13161)_ = 14.6, ŋ^2^ < 0.01, *p* < 0.01) as well as *sex × race* interaction (F_(3,13161)_ = 2619.0, ŋ^2^ = 0.01, *p* < 0.01).

### 3.2. Pace Change in Men and Women in 10-km and Marathon

Regarding pace change (Figure 2), the two-way ANOVA showed significant main effects of *sex*(F_(3,25139)_ = 129.6, ŋ^2^ < 0.01, *p* < 0.01), *race*(F_(3,25139)_ = 3717.2, ŋ^2^ = 0.13, *p* < 0.01) as well as *sex × race* interaction (F_(3,25139)_ = 149.3, ŋ^2^ = 0.01, *p* < 0.01).

### 3.3. Age Group Pace Change in 10-km and Marathon

For women (Figure 3, upper panel), the two-way ANOVA showed significant main effects of *age*(F_(17,11962)_ = 359.3, ŋ^2^ = 0.01, *p* < 0.01), *race*(F_(17,11962)_ = 441.1, ŋ^2^ = 0.04, *p* < 0.01) as well as *age × race* interaction (F_(17,11962)_ = 189.3, ŋ^2^ < 0.01, *p* < 0.01), whereas for men (Figure 3, lower panel), the two-way ANOVA showed significant main effects of *age*(F_(17,13145)_ = 3.9, ŋ^2^ < 0.01, *p* < 0.01), *race*(F_(17,13145)_ = 1678.2, ŋ^2^ = 0.11, *p* < 0.01) as well as *age × race* interaction (F_(17,13145)_ = 4.0, ŋ^2^ < 0.01, *p* < 0.01).

## 4. Discussion

The main aim of this study was to assess and compare pacing between 10-km runners and marathoners, in regards to their sex and age. The hypothesis, that 10-km runners would show more even pacing than marathoners, particularly women and middle age runners, was partially confirmed.

Contrary to the previous results obtained on elite 10-km runners, where an even or negative pacing profile was observed [1,4], recreational runners (both men and women) showed a positive pacing profile (Figure 1). However, when compared to the marathoners, the pace slowing that occurred in the second half of the 10-km race was significantly less. It seems that running performance does not affect pacing in the 10-km, as it does in the marathon [11,26]. In our study, marathoners achieved a faster speed than the 10-km runners (Table 1), thus we can assume that the 10-km runners were mostly beginners in comparison to the marathoners. Similar results were previously obtained for recreational half-marathoners [10,22], where plummet in pace was less prominent in recreational half-marathoners than the marathoners. Therefore, we can assume that the pacing difference between 10-km race and marathon was not related to the runners’ performance level, but could be attributed to the increased fatigue when participating in the longer distance race [22]. 

### 4.1. Pacing in 10-km and Marathon

Finally, note that time (expressed as minutes and seconds) per kilometre was utilized as a unit of speed. Similar studies have often used meters per second or kilometres per hour [10,11] as more common units of speed, particularly in the field of sports science. However, practitioners (e.g., coaches, runners of all levels, as well as some sports scientists) regularly use time per kilometre as a very comprehensive practical tool when assessing and comparing someone’s running speed or pace maintenance (see Materials and Methods for additional information). 

### 4.2. Pace Change in Men and Women in 10-km and Marathon

When women and men were compared, no differences in pace change were obtained in the 10-km race (Figure 2). In contrast to the 10-km race, the marathon women had a significantly lower pace change in comparison to the marathon men. When we relate this finding to the previous studies comparing pacing in recreational half-marathoners and marathoners [10,22], it appears that in long-distance races shorter than a marathon, the men’s pacing strategy is equally good as the women’s one, potentially event slightly better (Figure 2). That would confirm the previously observed hypothesis that women have an advantage in pacing over men only in distances equal to or longer than marathons [10]. The obtained findings can possibly diminish the previously reported psychological effect on men’s pacing (i.e., a riskier and faster start by men, due to greater competitiveness [14,15,27]). It can be assumed that less variation in pacing, as observed in recreational female marathoners, was due to a better fat utilization to obtain energy [28], rather than burning glycogen stored in muscles [12].

### 4.3. Age Group Pace Change in 10-km and Marathon

In all age groups (in both women and men), the 10-km runners showed a lower pace change in comparison to the marathon runners (Figure 3). It appears that both the youngest and the oldest marathoners change pace more than other age groups (Figure 3, lower panel), which is also confirmed in the 2017 Vienna marathon and half-marathon [29]. Younger, less experienced marathoners might encounter an inadequate control mechanism of pacing by altering pace often, which in turn might induce an excess of fatigue. The control mechanism of pacing depends on the information of the endpoint and race duration, an inherent clock setting scalar timing, and the knowledge of pacing from previously finished races [30]. Younger inexperienced runners could lack this pacing template in the brain, hence they cannot control pace during prolonged activities, such as a marathon. On the other hand, elderly men spend more time running (i.e., run slowly). As a consequence, fatigue-induced change in pace is more likely to occur, regardless of the well-developed pacing template [29]. Similar results are obtained in women, whereas the youngest and oldest marathoners change pace the most (Figure 3, upper panel). This phenomenon is, however, less pronounced compared to men due to previously explained sex differences (see preceding paragraph).

### 4.4. Limitations

Several limitations of this study should be noted. First, it was impossible to include the half-marathon mid-race split in this study, since the organizers usually don’t provide a split at 10.550 km. Also, this split is not quite popular among runners (both elite and recreational). Second, additional splits in the 10-km race would provide better insight into the runners’ pacing strategies. Unfortunately, almost no mass participation events provide these splits in races shorter than half-marathon. Third, this study has assessed only one event in four consecutive years, thus limiting the potential generalization of the obtained findings. On the other hand, this allows a greater sample of participants. Finally, combining all runners older than 65 into the one age group (since there is a limited number of runners older than 65), limits our knowledge about pacing in older 10-km and marathon runners.

### 4.5. Practical Applications

Based on these findings, strength and conditioning coaches (e.g., running coaches) should adopt different training programmes for marathoners, in comparison to the participants in the shorter long-distance events. Particular emphasis should be placed on individualized training plans for beginners, with the purpose of achieving an even or negative pacing profile (or at least to reduce plummet in the speed in the second half of the race). For example, recreational runners could run several 10-km races or half-marathons with the goal of achieving an even or negative pacing profile, before they attempt to run a marathon. This pacing strategy might aid athletes to have a faster race time, decrease the risk of musculoskeletal injuries, and enhance the enjoyment of endurance running. 

Finally, a particular novelty of this study was the use of minutes per kilometre as a unit of speed, as a very “practitioner friendly” and quite comprehensive tool when assessing and comparing someone’s running speed or pace maintenance (see Methods for additional information). Similar studies could utilize this measurement of speed more often, thus providing more comparable findings. 

## 5. Conclusions

In conclusion, both 10-km runners and marathoners showed positive pacing strategies. Moreover, women are less likely to slow in the marathon, however not in shorter long-distance events. Finally, pace changing is more prominent in youngest and oldest marathoners comparing to the other age groups. 

## Figures and Tables

**Figure 1 medicina-57-00389-f001:**
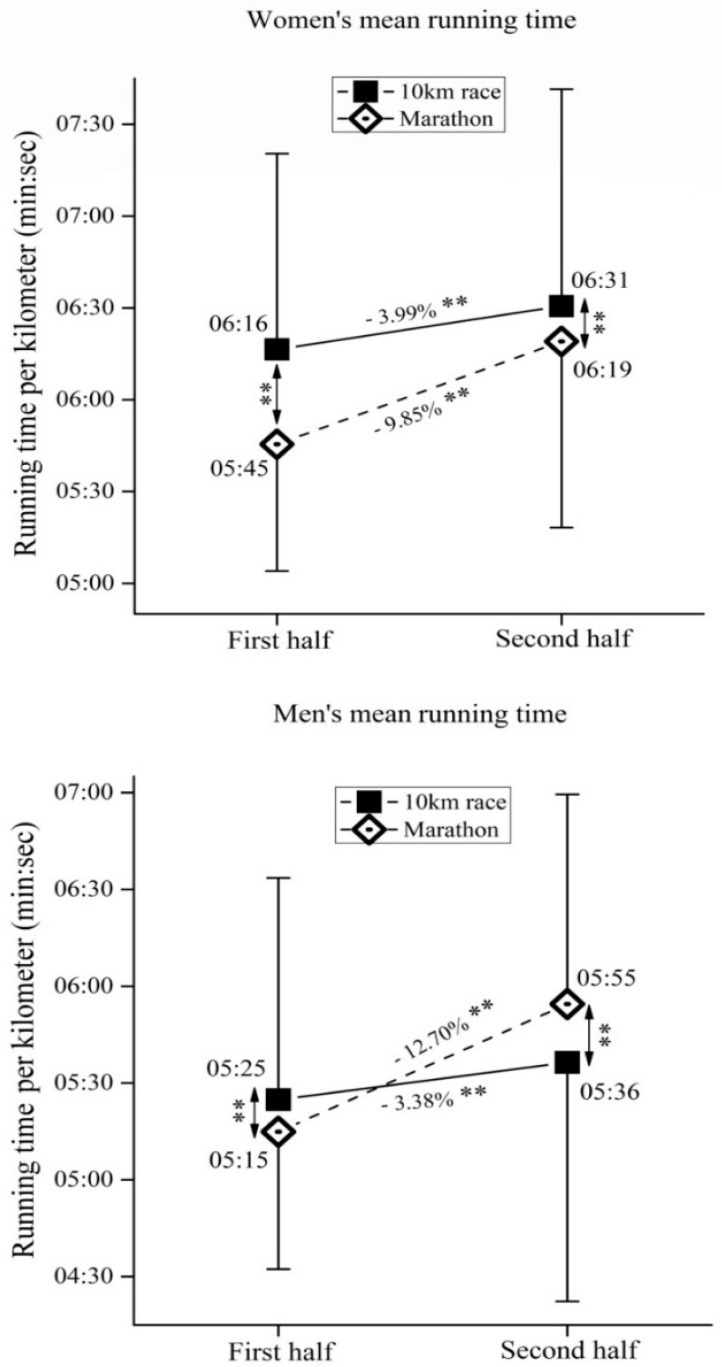
Women’s (upper panel) and men’s (lower panel) running time in the first and second half of 10-km race and marathon. Data showed as mean ± standard deviation. **—Significant differences at *p* < 0.01.

**Figure 2 medicina-57-00389-f002:**
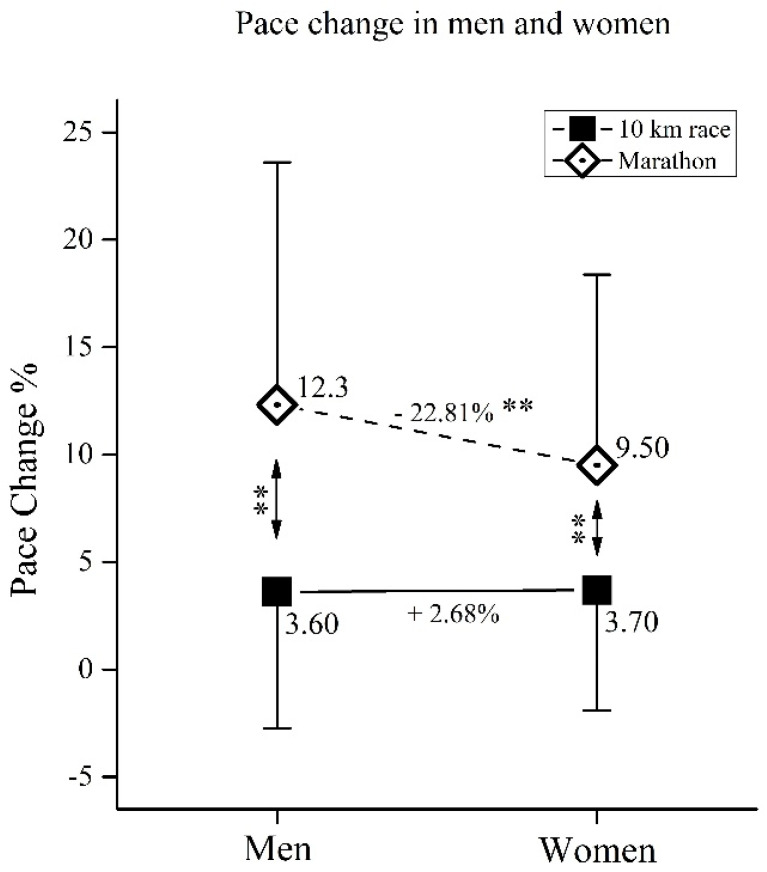
Pace change in 10-km race and marathon for women and men. Data showed as mean ± standard deviation. **—Significant differences at *p* < 0.01.

**Figure 3 medicina-57-00389-f003:**
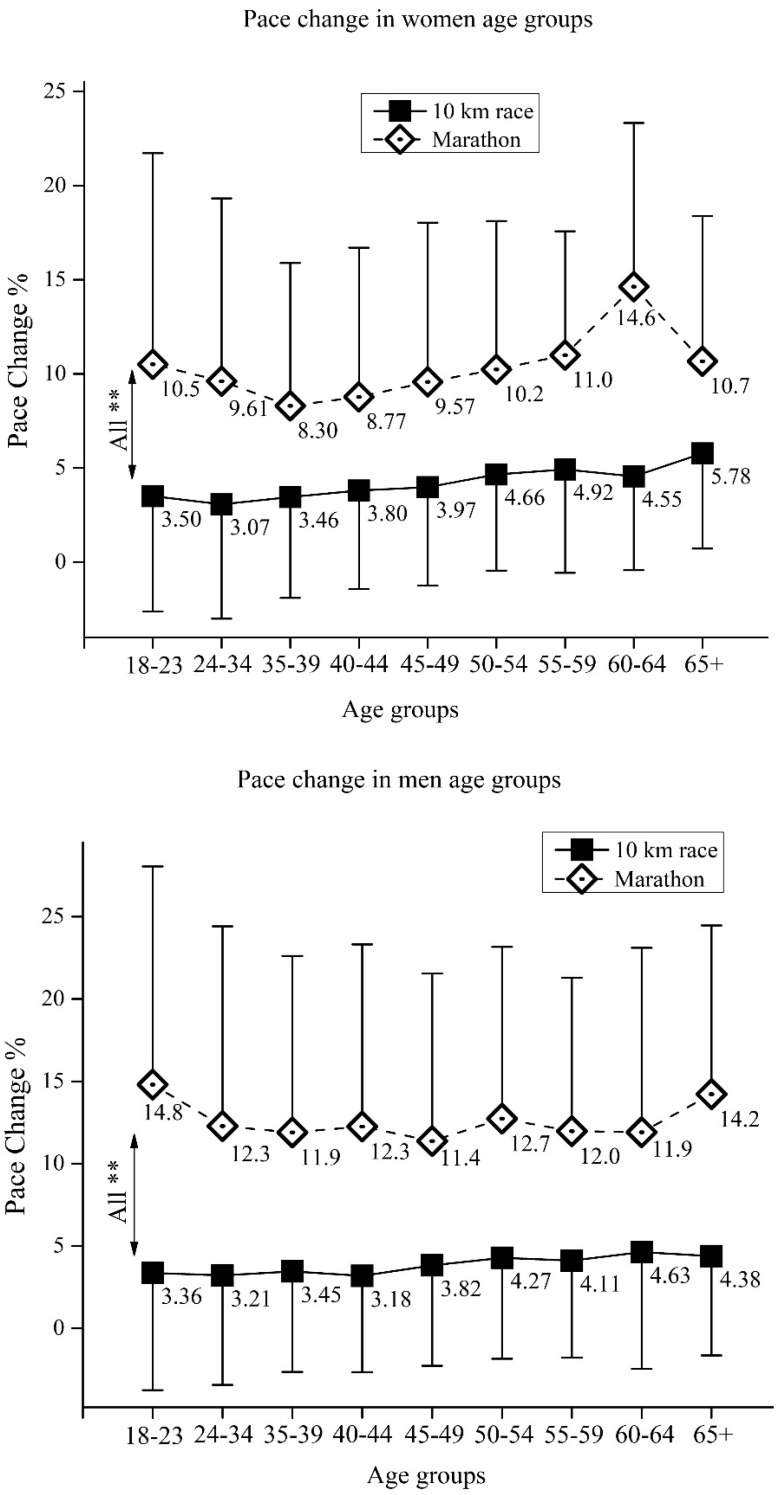
Pace change in 10-km race and marathon for women’s (upper panel) and men’s (lower panel) age groups. Data showed as mean ± standard deviation. **—Significant differences at *p* < 0.01.

**Table 1 medicina-57-00389-t001:** Speed indicators (in min/km) of 10-km and marathon runners showed as mean ± standard deviation.

		Women (*n*_10-km_ = 9932; *n*_marathon_ = 2048)				Men (*n*_10-km_ = 6383; *n*_marathon_ = 6780)			
		10-km Race (min/km)		Marathon (min/km)		10-km Race (min/km)		Marathon (min/km)	
		Mean	SD	Mean	SD	Mean	SD	Mean	SD
Age: 18–23 *n* = 1589	First half	6:03.2	1:05.9	5:50.0	0:38.9	5:03.1	1:06.3	5:17.8	0:42.4
	Second half	6:16.0	1:12.9	6:27.7	1:04.9	5:12.7	1:10.0	6:06.0	1:11.6
	Total	6:09.6	1:08.5	6:08.9	0:49.4	5:07.9	1:07.1	5:41.9	0:54.3
Age: 24–34 *n* = 7777	First half	6:07.7	1:02.7	5:41.2	0:40.7	5:13.2	1:06.0	5:10.0	0:42.5
	Second half	6:19.1	1:09.6	6:15.0	1:02.9	5:22.9	1:10.7	5:49.1	1:07.1
	Total	6:13.4	1:05.2	5:58.1	0:49.8	5:18.1	1:07.4	5:29.6	0:52.4
Age: 34–39 *n* = 3695	First half	6:14.7	1:02.2	5:36.1	0:37.5	5:17.7	1:06.3	5:09.9	0:43.7
	Second half	6:27.7	1:08.2	6:04.4	0:52.4	5:28.4	1:10.1	5:47.5	1:04.0
	Total	6:21.2	1:04.4	5:50.2	0:43.4	5:23.1	1:07.4	5:28.7	0:51.9
Age: 40–44 *n* = 3784	First half	6:17.9	1:01.1	5:46.3	0:42.6	5:19.3	1:05.2	5:12.5	0:39.4
	Second half	6:32.3	1:07.1	6:17.4	0:59.5	5:29.3	1:09.5	5:51.4	1:00.4
	Total	6:25.1	1:03.3	6:01.8	0:49.6	5:24.3	1:06.6	5:32.0	0:47.6
Age: 45–49 *n* = 3514	First half	6:18.7	1:01.4	5:47.7	0:37.4	5:29.0	1:06.4	5:14.6	0:40.1
	Second half	6:34.0	1:08.5	6:21.5	0:55.4	5:41.5	1:11.9	5:51.0	0:59.8
	Total	6:26.4	1:04.2	6:04.6	0:44.6	5:35.3	1:08.4	5:32.8	0:47.9
Age: 50–54 *n* = 2342	First half	6:23.4	1:04.8	5:52.8	0:43.1	5:34.1	1:07.2	5:23.2	0:41.6
	Second half	6:41.4	1:11.5	6:29.3	0:58.5	5:48.3	1:13.0	6:05.0	1:02.4
	Total	6:32.4	1:07.4	6:11.0	0:49.2	5:41.2	1:09.3	5:44.1	0:49.9
Age: 55–59 *n* = 1256	First half	6:45.1	1:06.2	6:00.3	0:39.7	5:43.7	1:06.4	5:23.6	0:39.1
	Second half	7:05.0	1:12.7	6:40.1	0:51.9	5:58.0	1:13.7	6:03.0	0:58.0
	Total	6:55.1	1:08.5	6:20.2	0:44.5	5:50.8	1:09.4	5:43.3	0:46.7
Age: 60–64 *n* = 646	First half	6:52.6	1:09.2	6:29.6	0:50.1	5:53.3	1:17.4	5:37.0	0:47.1
	Second half	7:10.9	1:11.8	7:26.8	1:07.5	6:09.4	1:22.8	6:17.3	1:05.3
	Total	7:01.7	1:09.7	6:58.2	0:56.8	6:01.3	1:19.0	5:57.2	0:53.5
Age: 65+ *n* = 540	First half	7:03.6	1:03.8	6:44.0	0:41.5	6:18.7	1:09.1	6:02.3	0:44.0
	Second half	7:28.2	1:11.2	7:27.9	1:02.7	6:35.4	1:16.1	6:54.7	1:06.9
	Total	7:15.9	1:06.7	7:06.0	0:50.5	6:27.1	1:11.6	6:28.5	0:53.1

*n* = number of participants, SD = standard deviation of data, min/km = minutes per kilometer.

## Data Availability

BMW Oslo Maraton. Available online: https://oslomaraton.no/ (accessed on 25April 2019).

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
