# Peer review of "Pacing in Long-Distance Running: Sex and Age Differences in 10-km Race and Marathon"

_medicina, 2021, doi:10.3390/medicina57040389_

Round 1

Reviewer 1 Report

Thank you for the opportunity to review an article written at such a high level. However, I have one request, in the Meterial and Methods section, please write how the authors obtained the data? Who made this data available, where did they get the statistical information for further calculations. 

Author Response

Reviewer: 1

Comments to the Author

Thank you for the opportunity to review an article written at such a high level. However, I have one request, in the Material and Methods section, please write how the authors obtained the data? Who made this data available, where did they get the statistical information for further calculations?

Answer: We are very thankful for these kind words and concerning comment from the expert reviewer. To answer all the raised questions, we have added a new subsection under Materials & Methods (2.2. Data Acquisition).

Reviewer 2 Report

General comments

By and large, this is a useful, albeit largely descriptive study of the similarities and differences in pacing between 10-km runners and marathoners as a function of both sex and age. The large number of participants and data included is an important asset of the study, which renders it worthy of publication.

The results cast some light on the mechanisms underlying pacing, but are particularly of relevance to running coaches and strength and conditioning coaches.

The main motivation for the present study is that “no study utilizing mass-participation events data has ever directly compared pacing in a 10-km race, with other long-distance races”. However, this is not a particularly strong motivation: why is such a study warranted from a theoretical and/or a practical point of view? (After all, there are many issues that have not been investigated before by scientists and for very good reasons, namely because they are not sufficiently interesting or important.) The rationale for the study could be strengthened by providing more substantive reasons for the comparison of pacing behavior between 10-km races and marathons. For instance, the argument could be advanced that such a comparison could shed light on the relative importance of the mechanisms underlying pacing behavior that are discussed in the paper. In addition, the argument could be made that such a comparison could also shed light on the training requirements for recreational runners vis-à-vis more experienced and proficient runners.

The quality of the English writing is acceptable in the sense that the text is generally comprehensible. At the same time, however, the text contains some imperfections, which hopefully will be eliminated in the copy-editing process. The Detailed comments below consist for the most part of detailed suggestions for improving the text. However, also once these suggestions have been incorporated, the manuscript will still require careful copy-editing.

Detailed comments

Page 1, Abstract, line 12-15: Rewrite first two sentences as follows: “The recent availability of data from mass-participation running events has allowed researchers to examine pacing from the perspective of non-elite distance runners. Based on an extensive analysis of the literature, we concluded that no study utilizing mass-participation events data has ever directly compared pacing in the 10-km race, with other long-distance races.”

Page 1, Abstract, line 19: Rewrite sentence as follows: “Both 10-km runners and marathoners showed positive pacing strategies.” (i.e. eliminating the mix-up of “both and” and “as well as”)

Page 1, line 31: “the risk of” instead of “the risk from”.

Page 1, line 34: “an end-spurt” instead of “the end-spurt”.

Page 1, line 37: “throughout” instead of “through”.

Page 1, line 38: delete “the” before “long-distance running”.

Page 1, line 40: “only involved” instead of “have been conducted on”.

Page 1, line 42: “to examine” instead of “to look into” (i.e. less colloquial).

Page 2, line 48-49: “in men than in women” instead of “in men”.

Page 2, line 54: “small samples” instead of “small sample”.

Page 2, line 62: “to better understand” instead of “in understanding”.

Page 2, line 63: Rework sentence into: “This might help runners to enjoy running more as well as to achieve better results.”

Page 2, line 64: “methodological approaches” instead of “methodology approaches”.

Page 2, line 65: “in mass-participation events” instead of “in the different mass-participation events”.

Page 2, line 68: “methodological approach” instead of “methodology approach”.

Page 2, line 78: “the observed” instead of “obtained”.

Page 2, line 83: “Based on an extensive analysis of the literature,” instead of “Based on the previous extensive literature analysis,”.

Page 2, line 94-95: Rework sentence into: “The Oslo marathon was chosen as an officially certified race because it was held on a rather flat track (elevation difference 60 m).” (At least that is how I understand the sentence.)

Page 2, line 95: Insert “the” before “Berlin marathon”.

Page 2, line 98: delete “the part of”.

Page 3, line 99: insert “the” before “Oslo marathon”.

Page 3, line 115: “Excel” instead of “excel”.

Page 3, line 120: “of all runners” instead of “for all runners”.

Page 4, line 157: “the age x race interaction” instead of “their interaction age x race”.

Page 4, line 165: “both 10-km runners and marathoners” instead of “both 10-km runners, as well as marathoners”.

Page 5, line 206: “the pace slowing” instead of “slowing in the pace”.

Page 5, line 208-209: “a faster speed than the 10-km runners” instead of “a faster speed in regards to the 10-km runners”.

Page 5, line 211: “obtained for” instead of “obtained on”.

Page 5, line 212: “than” instead of “in comparison to”.

Page 5, line 212: “could be attributed” instead of “vastly attributed”.

Page 7, line 236: “In contrast to” instead of “Conversely to”.

Page 7, line 237: “marathoners [10,22], it appears” instead of “marathoners [10,22]. It appears”.

Page 7, line 241: “That would confirm” instead of “That can confirm”.

Page 9, line 267: “less pronounced compared to men” instead of “less emphasized comparing to men”.

Page 9, line 271: “in this study” instead of “into this study”.

Page 9, line 273: “insight” instead of “inside”.

Page 9, line 284: “should be placed on” instead of “should be placed”.

Reviewer 3 Report

The aim of the study entitled “Pacing in long-distance running: sex and age differences in 10-km race and marathon” was to assess and compare pacing between 10-km runners and marathoners, in regard to their sex and age. The topic is interesting, and the paper is very well written in form and content. Considering its practical applications, I would recommend it for publication once the authors address a few concerns.

Specific comments

Abstract

Abstract, methods: add detail about the statistical model used for the analysis of the data.

Abstract, results: add differences between age groups.

Introduction

This section needs to be little bit shortened to appear more focused.

Methods (Participants): add the ability level of the participants, e.g., athletes, well-trained, recreational…

How many males and how many females there were in the two groups of runners? This information should be added in the table 1.

Line 119: a citation should be added about the “runners friendly” measurement of speed.

Discussion

Line 205-213: I suggest removing the summary of the results.

Line 227: …fitness level…What you mean? Please clarify

Author Response

Reviewer: 2

The aim of the study entitled “Pacing in long-distance running: sex and age differences in 10-km race and marathon” was to assess and compare pacing between 10-km runners and marathoners, in regard to their sex and age. The topic is interesting, and the paper is very well written in form and content. Considering its practical applications, I would recommend it for publication once the authors address a few concerns.

 Answer: We are very thankful for these kind comments from the expert reviewer.

Specific comments

 Abstract

Abstract, methods: add detail about the statistical model used for the analysis of the data.

 Answer: We are very thankful for this suggestion and we have added this in brief (please note that abstract is limited with words, so we had to be concise).

Abstract, results: add differences between age groups.

Answer: We are very thankful for this suggestion. Somehow we missed to add this information to the abstract in the first place. Now it is added.

Introduction

This section needs to be little bit shortened to appear more focused.

 Answer: We are very thankful for these kind comments from the expert reviewer. Some changes we made in this regards. Please see the new version of the manuscript, in particular, text in red.

Methods (Participants): add the ability level of the participants, e.g., athletes, well-trained, recreational…

Answer: We are very thankful for these kind comments from the expert reviewer. We have added this information to the manuscript.

How many males and how many females there were in the two groups of runners? This information should be added in the table 1.

Answer: We are very thankful for raising this issue. We have added this information to the table 1. Furthermore, we have added more information

Line 119: a citation should be added about the “runners friendly” measurement of speed.

 Answer: We are thankful for this kind comment. However, we were unable to provide any citation suitable for a scientific paper. On the other hand, we believe that we explained in following sentences why this is “runners friendly” measurement of speed.

Discussion

Line 205-213: I suggest removing the summary of the results.

Answer: We are thankful for this comment. We have removed this summary as proposed.

Line 227: …fitness level…What you mean? Please clarify

Answer: We are thankful for this suggestion. We have changed that sentence to be more understandable.